# Preparation of DSPE-PEG-cRGD Modified Cationic Liposomes for Delivery of OC-2 shRNA and The Antitumor Effects on Breast Cancer

**DOI:** 10.3390/pharmaceutics14102157

**Published:** 2022-10-10

**Authors:** Chunyan Liu, Wenli Zhao, Ligang Zhang, Huamin Sun, Xi Chen, Ning Deng

**Affiliations:** Guangdong Province Engineering Research Center for Antibody Drug and Immunoassay, Department of Biology, Jinan University, Guangzhou 510000, China

**Keywords:** OC-2, breast cancer, cRGD, cationic liposomes, shRNA

## Abstract

Cationic liposome delivery of interfering RNA (shRNA) plays an important role in tumor therapy. The cyclic Arg-Gly-Asp (cRGD) modified cationic liposomes (cRGD-CL) were designed for targeted delivery of ONECUT2 (OC-2) shRNA (pshOC-2) to breast cancer cells. The characterization analysis of cationic liposome showed that the prepared cRGD-CL/pshOC-2 lipoplexes had uniform particle size (150 ± 1.02 nm), moderate zeta potential (19.8 ± 0.249 mV) and high encapsulation efficiency (up to 96%). The results of flow cytometer showed that the introduction of cRGD could significantly promote the liposomes targeting tumor cells. In MCF-7 cells, the pshOC-2 could down-regulate expression of OC-2 and result in cell apoptosis, inhibition of the wound healing, migration and cell colony formation, in which the signal pathways of Bcl-xL, Bcl-2 were inhibited and the signal pathways of Bax and Cleaved Caspase-3 were promoted. In MCF-7 xenograft mice, intravenous administration of cRGD-CL/pshOC-2 lipoplexes could effectively reduce the expression of OC-2 in tumors and result in apparently antitumor effects, which suggested that the lipoplexes might be deeply penetrated into tumor through receptor-mediated transcytosis. The results revealed that the cationic liposome (cRGD-CL) was an effective delivery system for OC-2 shRNA, which might be an effective therapeutic candidate for breast cancer.

## 1. Introduction

Breast cancer is the second leading cause of mortality among women, with the highest incidence worldwide in 2020 [1,2]. Endocrine therapy and chemotherapy are the primary treatments for breast cancer, but their therapeutic efficacy is limited and occasionally unsatisfactory due to high tumor heterogeneity and drug resistance [3,4]. Gene therapy is one of the most promising and effective therapeutic approaches in breast cancer treatment, which has attracted widespread attention and expectations, and the discovery of new therapeutic targets and delivery systems are of great importance [5].

Carcinoma susceptibility genes and abnormally expressed genes are potential targets for breast cancer gene therapy [5]. OC-2, a newly discovered transcription factor that belongs to ONECUT family, has been demonstrated to be carcinogenic potent in a variety of tumors [6], such as gastric cancer [7,8], ovarian cancer [9,10], prostate cancer [11,12], and colorectal cancer [13]. Interestingly, Shen et al. reported that the downregulation of OC-2 mediated by chemotherapy-induced extracellular vesicle (EV) miRNA was thought to promote the stemness and chemoresistance of breast cancer, though the overexpression of OC-2 dramatically facilitated the growth of xenograft mammary tumors [14]. The investigation provides a potential strategy to enhance chemotherapeutic efficacy by blocking the EV miRNA-OC-2 axis, but the role of OC-2 in breast cancer development and gene therapy has not been clarified. However, the chemoresistance mediated by downregulation of OC-2 seems to obscure the potential value of OC-2 as a gene therapeutic target for breast cancer. Accordingly, it is essential to explore whether interference of OC-2 expression can be used as an approach for the treatment of breast cancer without increasing the risk of malignant transformation.

The transfection efficiency of cationic liposomes is lower than that of viral vectors, but they are considered to be more valuable gene vectors for research because of their excellent biosafety, transfection efficiency and biocompatibility; cationic liposomes can be loaded with different nucleic acid molecules, plasmids and siRNAs are more common [15]. Compared with shRNA, the electrostatic interaction between siRNA and liposomes is not controlled, which may lead to excessive particle size and poor stability. Another potential problem is the low siRNA encapsulation rate, which exposes siRNA to potential degradation by enzymes before entering cells [16]. Cationic liposomes (CLs) are positively charged subspherical vesicles, mainly composed of cationic liposomes and neutral. The positive charge on the surface of cationic liposomes promotes contact with negatively charged cell membranes [17], but on the other hand, they also bind to serum proteins and form larger polymers that do not easily enter the cells, so it is imperative to modify the surface of cationic liposomes, and the common way is PEG modification, which also reduces the toxicity of cationic liposomes [18].

The nanocarriers are crucial to achieving the intracorporal therapeutic efficacy, and targeted nanoparticles with ligand modification are expected to maintain or enhance the therapeutic outcome. As integrin αvβ3 is upregulated in tumor and angiogenic endothelial cells [19,20], it is widely used in the recognition of cRGD tripeptide in nucleic acid delivery systems [19]. Studies have found that active-targeting nanoparticles modified with cRGD ligand could enhance their binding to αvβ3 overexpressing cells and initiate the process of receptor-mediated transcytosis [20,21,22].

Here, we designed and prepared cRGD modified cationic liposomes (cRGD-CL) for the delivery of OC-2 shRNA (pshOC-2) to investigate the effect of OC-2 interference on breast cancer survival, migration and metastasis. The characterization and delivery performance of cRGD-CL including tumor targeting and transfection efficiency were revealed. The effects of cRGD-CL/pshOC-2 liposome-mediated OC-2 knockdown on colony formation, wound healing and migration of MCF-7 cells were investigated, and their antitumor activity, biodistribution and biosafety were evaluated in mice. The purpose was to reveal that the reduction in tumor growth was attributed to the pro-apoptotic effect of OC-2 knockdown mediated by cRGD-CL/pshOC-2 lipoplexes and the mechanism of transportation for its genetic cargo to tumor cells via transcytosis [21,22].

## 2. Materials and Methods

### 2.1. Materials

(2,3-Dioleoyloxy-propyl)-trimethylammonium-chloride (DOTAP) was purchased from Corden Pharma (#LP-R4-117, Liestal, Switzerland), DMG-PEG 2000 was purchased from Avanti Polar Lipids (#880151P, Alabaster, AL, USA). DSPE-PEG-cRGD was purchased from Ruixibio (#R-9995, Shanxi, China). Cholesterol was purchased from Macklin (#C804519, Shanghai, China). Lipofectamine™ 2000 was purchased from Invitrogen (#11668-019, Carlsbad, CA, USA). Plasmid DNA encoding green fluorescent protein (GFP) and short hairpin RNA targeting *OC-2* was purchased from GenePharma (Shanghai, China): Sense: 5′-GCCAGCTGGAAGAAATCAACA -3′, anti-sense: 5′-TGTTGATTTCTTCCAGCTGGC-3′.

### 2.2. Cell Culture

Human breast cancer cell lines MCF-7 and human embryonic kidney cells (HEK293T) were obtained from the Shanghai Institute of Biochemistry and Cell Biology, Chinese Academy of Sciences. All the cells were cultured in Dulbecco’s modified Eagle’s medium (DMEM, Gibco, Waltham, MA, USA) supplemented with 10% fetal bovine serum (FBS, Biological Industries, Kibbutz Beit Haemek, Israel) and 100 U/mL penicillin/streptomycin (Gibco, Waltham, MA, USA). All the cells were incubated in a 5% CO_2_ incubator at 37 °C.

### 2.3. Animal

Female 4-week-old BALB/c nude mice were purchased from Beijing Huafukang Biological Co. Ltd. (Beijing, China). The BALB/c nude mice were fed in a specific pathogen-free (SPF) environment with the temperature maintained at 25–27 °C. All the animals used in the experiment were treated humanely in accordance with the guidelines of the Institutional Animal Care and Use Committee (IACUC) in Jinan University.

### 2.4. Preparation of cRGD-CL and CL

Bare liposomes were prepared by hydration of freeze-dried matrix (HFDM) method [23]. Lipids of cRGD-CL (DOTAP, cholesterol, DMG-PEG2k, DSPE-PEG2k-cRGD, 50:42:2:6, molar ratios) or CL (DOTAP, cholesterol, DMG-PEG 2k, 50:42:8, molar ratios) were dissolved in tert-butanol and mixed with equal volume of filtered sucrose solution (15 mg/mL), respectively. In terms of FITC or DiR labeling liposomes, 0.2% molar ratio FITC or DiR Iodide was added into the mixture. Then, they were snap-frozen for 12 h and freeze-dried for 72 h (Christ ALPHA 2-4 LD plus) at a condensing temperature of −89 °C and pressure of less than 0.1 mbar. The lyophilized powder was dissolved in sterile water and gently shaken to form liposomes with polydispersed hydrodynamic diameter. To narrow down their size distribution, liposomes were extruded through 400 nm and 200 nm polycarbonate membrane 11 times in LiposoEasy LE-1 (MoGe Machinery, Shanghai, China), or treated with ultrasound in cold water for 30 min.

### 2.5. Characterization of cRGD-CL

The hydrodynamic diameter, polydispersity (PDI) and zeta potential of the optimized blank liposomes (cRGD-CL or CL) were measured using a Zetasizer Nano ZS90 (Malvern Instruments, Malvern, UK). The stability of cRGD-CL was evaluated by measuring the size changes during the incubation with H_2_O, PBS and 10%FBS+DMEM at room temperature for 48 h. The morphology of cRGD-CL negatively stained with 1% phosphotungstic acid was observed by transmission electron microscopy (TEM, FEI).

### 2.6. Nucleic Acid Encapsulation Efficiency

cRGD-CL/pshOC-2 lipoplexes at the N/P ratios of 0.5, 1, 2, 3 and 4 were subject to 1% Agarose retardation assay. The pshOC-2 encapsulation efficiency of CL and cRGD-CL at the N/P ratios of 2, 4, 6, 8 and 10) was measured using Picogreen dsDNA quantitation reagent (Yeasen, Shanghai, China). The fluorescein intensity of free pDNA (excitation wavelength: 480 nm, emission wavelength: 520 nm) before and after rupture by 0.1% TritonX-100 was detected by a microplate reader (BioTek Instruments, CA, USA). Encapsulation efficiency = 100% × [Fluorescein Intensity (ruptured after)—Fluorescein Intensity (ruptured before)]/Fluorescein Intensity (ruptured after).

In addition, the protective effect of cRGD-CL on pshOC-2 (0.3 μg) was determined by DNase I enzymatic hydrolysis assay. The cRGD-CL/pshOC-2 lipoplexes with N/P ratios of 2, 3 and 4 were interacted with DNase I for 10 min and then DNase I was in-activated in a water bath at 60 °C for 5 min, and the liposomes were disrupted with 0.1% TritonX-100 to release their contents. Finally, all samples were subjected to 1% agarose gel electrophoresis.

### 2.7. Cytotoxicity Assay

MCF-7 cells (5 × 10^3^ cells per well) were seeded into 96-well plates and treated with 5–60 μg/mL cRGD-CL and CL for 24 h (n = 6). Lipofectamine 2000 (Lipo 2000, Invitrogen, Waltham, MA, USA) served as the positive control. In the dark, 10 μL CCK-8 reagent (GLPBIO) was added to each well and the optical density at 450 nm was read using a microplate reader.

### 2.8. Cellular Uptake

MCF-7 cells (8 × 10^4^ per well) were seeded into 12-well plates (n = 3) and cultured for 12 h. FITC-cRGD-CL or FITC-CL diluted in Opti-MEM (Gibco, Waltham, MA, USA) was added and incubated for 1, 2 and 4 h, respectively. After PBS washing, cells were collected to detect the total FITC intensity using flow cytometry (FCM, BD Accuri C6). For intracellular FITC intensity, the cell suspension of 4 h was stained with 0.4% Trypan blue (4:1, volume ratio) before the detection.

### 2.9. Intracellular Trafficking

MCF-7 cells (2 × 10^5^ per dish) were seeded into confocal dish (n = 3) and cultured for 12 h. FITC-cRGD-CL/pshOC-2 or FITC-CL/pshOC-2 lipoplexes diluted in Opti-MEM were added and incubated for 0.5, 1, 2 and 4 h, respectively. After PBS washing, cells were stained with 75 nM Lyso-Tracker Red/DMEM (#C1046, Beyotime, Shanghai, China) and fixed with 4% paraformaldehyde. Cell nuclei was stained with DAPI (#C1005, Beyotime, Shanghai, China) and the cells were imaged by confocal laser scanning microscopy (CLSM, Olympus, Tokyo, Japan).

### 2.10. Transfection Efficiency

293T and MCF-7 cells (8 × 10^4^ cells per well) were seeded into 12-well plates in DMEM with 10% FBS and incubated for 12 h at 37 °C. Liposomes (cRGD-CL or CL) and psh*OC-2*-GFP (1.5 μg) were diluted in 150 uL Opti-MEM, respectively, and mixed for incubation of 30 min to form a lipid-DNA complex (lipoplexes). Lipoplexes (300 μL) was then added into cells and transfected for 48 h in Opti-MEM. Finally, the GFP intensity was photographed by an inverted fluorescence microscope (Carl Zeiss), and quantified by FCM. To measure the transfection efficiency of cRGD-CL and CL under different N/P ratios, corresponding lipoplexes were transfected into MCF-7 cells for 48 h. Cells were collected and the cells expressing GFP were counted by FCM.

### 2.11. Western Blot

Cells and tumor tissues were lysed using RIPA buffer supplemented with protease inhibitors. Samples of 20 μg protein were subjected to SDS-PAGE and transferred to PVDF membranes. Membranes were blocked with 5% nonfat milk and incubated with primary antibodies against *OC-2* (Abcam, #ab28466), integrin αV (Cell Signaling Technology, CST, #4711), Cleaved Caspase-3 (CST, #9664), Bcl-2 (CST, #4223), Bcl-xL (CST, #2764), Bax (CST, #5023), PARP (CST, #9532), E-cadherin (CST, #3195), N-cadherin (CST, #13116) and Vimentin (Proteintech, #10366-1-AP) overnight at 4 °C, respectively. Following incubation with HRP-linked anti-rabbit IgG (Cell Signaling, #7074) secondary antibody, the membranes were exposed to ECL Western Blot Substrate (Millipore, MA, USA) for chemiluminescent detection. The protein relative expression was quantified with ImageJ software (NIH) and GAPDH (CST, #5174) served as a loading control.

### 2.12. In Vitro Antitumor Efficacy

#### 2.12.1. Colony Formation Assay

MCF-7 cells (200 cells per well) were seeded into 12-well plates and incubated for 12 h at 37 °C. Cells were treated with cRGD-CL/pshOC-2 lipoplexes (pshOC-2 1.5 μg/well, N/P ratios = 3 and 4) and bare liposomes cRGD-CL, respectively, and cultured in Opti-MEM for a week. Colonies were fixed with 4% paraformaldehyde, stained with crystal violet and photographed with a mobile phone (Apple, USA).

#### 2.12.2. Migration Assay

For wound healing assays, MCF-7 cells (2.8 × 10^5^ cells per well) were seeded into 6-well plates and treated with cRGD-CL/pshOC-2 lipoplexes (pshOC-2 = 1.5 μg/well, N/P ratios = 2, 3 and 4), bare liposomes cRGD-CL and CL, respectively, in Opti-MEM for 24 h. Then cells were wounded using pipette tips and cell migration was imaged at 0 h and 24 h using an inverted microscope (Carl Zeiss, Thuringia, Germany).

For Transwell migration assay, MCF-7 cells were treated with cRGD-CL/pshOC-2 lipoplexes (pshOC-2 = 1.5 μg/well, N/P = 4), cRGD-CL and CL for 24 h, respectively, and then seeded into the Transwell chamber (8 μm) at 8 × 10^4^ cells per chamber and incubated for another 24 h. Cells were fixed with 4% paraformaldehyde, stained with crystal violet, and imaged with an ordinary microscope (Olympus).

#### 2.12.3. Cell Apoptosis Assay

MCF-7 cells (8 × 10^4^ cells per well) were seeded into 12-well plates and treated with cRGD-CL/pshOC-2 (pshOC-2 = 1.5 μg/well, N/P ratios = 2, 3 and 4) for 48 h. The collected cells were stained with Annexin V-APC and 7-AAD (#E-CK-A218, Elabscience, Wuhan, China), followed by the detection of APC and PerCP/Cy5.5 channels in FCM.

### 2.13. In Vivo Antitumor Efficacy

MCF-7 cells (3 × 10^6^ cells/100 μL) were inoculated into the armpit of BALB/c nude mice. Tumor volume (V) and body weight were monitored every other day. V was calculated by the following formula: V = ab^2^/2, where a and b were the major and minor axes of the tumor measured using a vernier caliper. When tumors reached 100 mm^3^ in volume, mice were randomly divided into 5 groups (n = 5 per group). Lipoplexes (cRGD-CL or CL: pshOC2 = 4:1, molar ratio, 2 mg of pshOC2 per kg) were injected intravenously or intratumorally every 3 days for the first week and every other day for the remaining 13 days. PBS served as the negative control. On day 20, mice were sacrificed and tumors and major organs (heart, liver, spleen, lung and kidney) were harvested. Tumors were photographed and weighed, and the expression status of OC-2 and Cleaved Caspase3 were detected by Western blot. Subsequently, tumors and organs were fixed with 4% paraformaldehyde for 24 h, and paraffin-embedded sections were performed. Among them, the expression of OC-2, Ki67, CD31 and LYVE1 of tumor sections were analyzed by immunohistochemistry assay (IHC). Terminal deoxynucleotidyl transferase-mediated nick end labeling (TUNEL) technique was applied to determine apoptosis in tumors. This project was approved by the Laboratory Animal Ethics Committee of Jinan University (approval No. IACUC- 20201202-01).

#### 2.13.1. In Vivo Imaging

Mice (n = 5) with xenografts (about 200 mm^3^) in the armpit of the right lower limb were injected intravenously with 100 μL DiR lodide (Coolaber, Beijing, China) control, DiR-cRGD-CL (2 mg of pshOC2 per kg, the same below), DiR-CL, DiR-cRGD-CL/pshOC-2 and DiR-CL/pshOC-2, respectively. The mice were anesthetized at 3, 5, 8, 12 and 24 h using avertin, respectively. The fluorescence imaging was performed by IVIS Imaging System (IVIS Lumina Series III, MA, USA). At 24 h, the mice were sacrificed and the major organs and tumors were stripped for ex-vivo imaging (excitation wavelength: 740 nm, emission wavelength: 790 nm).

#### 2.13.2. Biosafety Evaluation

H&E staining was subject to detect the pathological changes of major organs. Total protein (TP)/albumin (ALB)/cholesterol (CHO)/lactate dehydrogenase (LDH) and blood urea nitrogen (BUN)/Creatinine (CREA) in the collected serum was assayed as indicators of liver and renal function, respectively.

### 2.14. Statistical Analysis

Data were presented as mean ± SD. All statistical analyses were performed using GraphPad Prism 6.0 software. The statistical significance was determined by one-way and two-way analyses of variance (ANOVA) and *t*-tests. Significance was determined according to *p* value, where * *p* < 0.05, ** *p* < 0.01, *** *p* < 0.001 and **** *p* < 0.0001.

## 3. Results

### 3.1. Preparation and Characterization of cRGD-CL

Hydration of freeze-dried matrix (HFDM) was reported to be a simple method of preparing liposomes suitable for body use without further control of size distribution [23]. However, the particle size of cRGD-CL prepared by HFDM was not uniform after rehydration (Figure 1A). Therefore, we attempted to optimize the distribution of particle size through ultrasound or extrusion. As shown in Figure 1B,C and Table 1, the hydrodynamic diameter distribution of cRGD-CL after extrusion was narrower than that after ultrasound. The average diameter was reduced from around 235 nm to 126 nm while PDI was reduced from 0.802 to 0.1, indicating that cRGD-CL was monodisperse after extrusion. Further, cRGD-CL was spherical with uniform distribution in TEM image (Figure 1D). Therefore, extrusion was used to optimize the particle size in the liposomal preparation.

The size and zeta potential of bare liposomes and the corresponding lipoplexes were measured subsequently. cRGD-CL (125 ± 0.918 nm, 35 ± 0.368 mV) was a little smaller and had a higher zeta potential than CL (146 ± 1.250 nm, 25 ± 1.970 mV) (Table 2), but after encapsulating the plasmid DNA (pDNA), there was only a slight difference in their zeta potential, indicating that they had the same ability of nonspecific binding with cell membrane through electrostatic interaction. In addition, the zeta potential of cRGD-CL lipoplexes under different N/P ratios (Figure 2A) was lower than the transfection reagent lipofectamine 2000 (Lipo2000), suggesting that they had lower cytotoxicity compared with Lipo2000.

The cytotoxicity assay of cRGD-CL in MCF-7 cells was conducted by CCK8 (cell counting *kit*-8). The Lipo2000 serves the positive control. The introduction of cRGD peptides appeared to increase the cytotoxicity of CL, but the cell viability of cRGD-CL was significantly higher than that of Lipo2000 even at a high concentration of 60 μg/mL, which indicated the cytotoxicity of cRGD-CL to MCF-7 cells was in an acceptable range (Figure 2B).

### 3.2. Stability and Nucleic Acid Loading Capacity of cRGD-CL

As a nanocarrier, cRGD-CL should be stable in different biological solutions. We measured the diameter and PDI of cRGD-CL after incubation with H_2_O, PBS and DMEM + 10% FBS within 48 h, respectively. In DMEM + 10% FBS, the size increased from 125 nm to 160 nm (Figure 2C) and PDI increased from 0.1 to 0.52 (Figure 2D), due to the uneven increase in particle size caused by the adsorption of serum proteins on liposomes. However, during the incubation within 48 h, no significant changes were observed on particle size and PDI of cRGD-CL under these 3 conditions, indicating that cRGD-CL could stay stable in biological solutions with different osmotic pressures. Moreover, the 1% agarose retardation assay showed that cRGD-CL could compress DNA completely when the N/P ratio was above 2 (Figure 2E). the encapsulation efficiency of lipoplexes at different N/P ratios was measured. the introduction of 6% cRGD-PEG-DSPE into CL greatly improved its nucleic acid encapsulation efficiency which achieved 96% at the N/P ratio of 4, DNase I was used to investigate whether cRGD-CL could protect pDNA from nuclease degradation at different N/P ratios (Figure 2F). There was little difference in the brightness of DNA bands between the groups treated with and without DNase I. Then lipoplexes were disrupted with 0.5% Triton X-100 to release the contents and found no difference in the brightness of DNA bands between them (Figure 2G). The results indicated that The DNA is wrapped inside by cRGD-CL nanoliposomal material to avoid degradation by DNase I.

### 3.3. Cell Uptake and Internalization of cRGD-CL

To evaluate the targeting ability of cRGD-CL to MCF-7 cells, the cellular uptake of fluorescein isothiocyanate (FITC) labeled liposomes was measured by flow cytometry (FCM). Compared with unmodified CL, the cRGD could significantly enhance the binding ability of liposomes in MCF-7 cells (Figure 3A). To detect the intensity of intracellular fluorescence, trypan blue was used to quench the extracellular fluorescence. After the fluorescence on the outer membrane was quenched, the total fluorescence intensity was somewhat attenuated in both groups. The cRGD-CL was effectively internalized in MCF-7 cells, whereas only 71% of MCF-7 cells treated with unmodified CL remained FITC positive (Figure 3B). Thus, cRGD-CL exhibited obvious advantages in internalization.

To further investigate the internalization and intracellular fate of FITC-cRGD-CL/pshOC-2 lipoplexes, we detected the endocytosis of lipoplexes by Confocal Laser Scanning Microscope (CLSM). In Figure 3C, cRGD-CL/pshOC-2 lipoplexes were taken up rapidly into MCF-7 cells and reached saturation at 2 h. The orange color represented the colocalization of lipoplexes and lysosomes. Lysosomes were the garbage dumps for cells to degrade the exogenous substances. However, at 4 h, the intensity of lysosomes labeling red fluorescence attenuated a lot, and simultaneously, the amount of lipoplexes colocated with lysosomes also decreased, suggesting that the integrity of the lysosome had been destroyed. Subsequently, the escaped lipoplexes tended to be distributed around the nucleus and might hence enter the nucleus, which was a key step for the exogenous DNA to express OC-2 shRNA. An appropriate nanocarrier should have the ability to escape from the entrapment of endosomes or lysosomes once endocytosed by cells. In a result, cRGD-CL/pshOC-2 lipoplexes could enter into cells rapidly through endocytosing and successfully escape from the endosomes or lysosomes to the nucleus (Figure 3D).

### 3.4. Transfection Effeciency Assay for cRGD-CL/pshOC-2

To investigate the effect of cRGD modification on transfection efficiency, 293T cells with low expression of integrin αV (ITGAV) and MCF-7 cells with high expression of ITGAV (Figure 4C) were transfected with cRGD-CL, CL and Lipo 2000, respectively. In 293T cells, the difference in transfection efficiency seemed to be related to the charge and structural organization of liposomes, while Lipo 2000 with the highest zeta potential and different composition acquired the highest transfection efficiency (83%) than CL (27%) and cRGD-CL (35%) with lower surface charge (Figure 4A,B). By contrast, the modification of cRGD ligands dramatically improved the transfection efficiency of CL in MCF-7 cells, while the transfection efficiency of non-targeted CL and Lipo 2000 was remarkably reduced.

The transfection efficiency of cRGD-CL/pshOC-2 at different N/P ratios was further detected in MCF-7 cells by FCM and Western blot. in Figure 4D,E, the transfection efficiency of cRGD-CL/pshOC-2 was reached 79.5% at the N/P ratio of 4. The Western Blot results showed that the OC2 expression was down-regulated with cRGD-CL/pshOC-2 about 70% at N/P = 4 (Figure 4F,G). In contrast, only a few cells expressed GFP after transfected with CL/pshOC-2 at any N/P ratios, suggesting that cRGD modification had an advantage in improving the transfection efficiency of liposomes in human breast cancer MCF-7 cells. Taken together, the results of Figure 4 showed that cRGD-CL could efficiently deliver pshOC-2 to MCF-7 cells, resulting in a significant OC-2 knockdown.

### 3.5. Suppressing of the Proliferation and Migration and Inducing Apoptosis in MCF-7 Cells with cRGD-CL/pshOC2 Lipoplexes

OC-2 has been reported to be highly related to the development of cancer [6]. To further investigate the role of OC-2 in MCF-7 cells proliferation and migration, a series of in vitro experiments of colony formation, wound healing, migration and apoptosis assays were conducted. The most obvious inhibition of cancer cell proliferation was observed in the N/P ratio of 4 of cRGD-CL/pshOC2 treated groups, while both the N/P ratio of 2 and 3 showed moderate effects (*p* < 0.05) (Figure 5A,E). The recovery area at 24 h resulted in obvious reduction in the N/P ratio of 4 of cRGD-CL/pshOC2 treatment, respectively, compared to PBS. The effects of PBS and free cRGD-CL were minimal (Figure 5B,F). Transwell assays were conducted to further demonstrate the anti-migration effect, which confirmed the inhibitory effect on the migration ability of cancer cells in the N/P ratio of 4 of cRGD-CL/pshOC2 treatment, respectively, compared to PBS (Figure 5C,G). The results showed that OC-2 knockdown significantly inhibited the ability of colony formation, wound healing and migration of MCF-7 cells. Since this efficacy was gradually strengthened followed by the enhancement of OC-2 knockdown. In particular, 35.06% of the cells were apoptotic followed by the treatment of cRGD-CL/pshOC-2 lipoplexes at an N/P ratio of 4 (Figure 5D,H), implying that OC-2 might play a critical role in tumor cell survival.

The expression of biomarkers related to apoptosis and epithelial-mesenchymal transition (EMT) was detected by Western blot. Compared with PBS and cRGD-CL/pshNC treatment groups, pro-apoptotic protein Bax and the cleaved fragments of PARP and Caspase-3 in cRGD-CL/pshOC-2 lipoplexes treatment group were up-regulated while pro-survival Bcl-2 and Bcl-xL were down-regulated simultaneously, which demonstrating that OC-2 knockdown definitely induced apoptosis in MCF-7 cells (Figure 5I,J). In accordance with the results of wound healing and migration assays, the expression of EMT-related proteins was decreased notably after transfection with cRGD-CL/pshOC-2 lipoplexes at N/P 4. E-cadherin, which was commonly thought to be inversely correlated with N-cadherin and deficient in tumor cells [24], was unexpectedly down-regulated in spite of the downregulation of N-cadherin and Vimentin. In addition, there was even no deficiency of E-cadherin in MCF-7 cells treated with PBS. However, in a recent study, E-cadherin has been reported to be critical for survival and metastasis in multiple models of breast cancer [25]. These results indicated that OC-2 knockdown was not conducive to the survival and metastasis of MCF-7 cells.

### 3.6. Suppressing for Tumor Growth in Xenograft Mouse Model Treated with pshOC-2 Lipoplexes

To evaluate the antitumor effect of cRGD-CL/pshOC-2 lipoplexes in vivo, the xenograft tumor BALB/c nude mice model was established. The MCF-7 cells (3.0 × 10^6^) were subcutaneously inoculated into the right lower extremity axilla of the mice. The cRGD-CL/pshOC-2 were administrated through intravenous injection (i.v.) and intratumoral injection (i.t.) in 2 days interval (Figure 6A). The CL/pshOC-2 was the control. The method of administration is also important to the efficacy of specific nanomedicine. In order to select an appropriate administration route for cRGD-CL/pshOC-2 lipoplexes, the effects of two administration routes on the antitumor efficacy of lipoplexes were compared at the same time. After 20 days of treatment, the tumor growth in i.v. injection groups were significantly inhibited relative to PBS control and i.t. injection group (Figure 6B–D). The poor therapeutic effect of i.t. injection group might be related to the complicated and dense tumor microenvironment (TME) where particles larger than 100 nm were basic immobility [26].

Considering antitumor activity, the cRGD-CL/ pshOC-2 lipoplexes (i.v.) treated mice showed the best inhibition rates in tumor weight (67.6%) and volume (66.2%), as well as in the expression of OC-2 (36.8%) than the other groups (Figure 6). Surprisingly, the non-targeted CL/pshOC-2 lipoplexes (i.v.) which showed poor transfection efficiency in vitro, also inhibited the expression of OC-2 (22.9%) and tumor growth (51.7% in tumor weight and 53.8% in volume) in vivo, and its antitumor effect was only marginally inferior to the targeted cRGD-CL/pshOC-2 lipoplexes (i.v.). Since the expression of OC-2 in cRGD-CL/pshOC-2 lipoplexes (i.v.) treatment group was obviously diminished, it was not possible to deduce that cRGD-CL/pshOC-2 lipoplexes were off-target. Liposomal drugs are coated with plasma proteins once injected intravenously into the physiological environment [27]. Dysopsonins are one such plasma proteins adsorbing on the surface of foreign particles, which endows the particles with the ability to evade capture by mononuclear phagocyte system (MPS) due to their few affinities for the cell surface [28,29,30]. We wondered whether specific dysopsonins were absorbed to the surface of CL/pshOC-2 lipoplexes after administration, endowing them with the capacity of tumor targeting and thus improving its intracorporal transfection efficiency [31,32]. Encouraged by the results above, we hypothesized that receptor-mediated transcytosis was responsible for their deep tumor penetration [20,26,33].

The results of Western blot showed that the downregulation of OC-2 evidently induced the up-regulation of the active cleaved fragment of Caspase-3 in tumor tissues, suggesting that OC-2 knockdown mediated by targeted or nontargeted CL/pshOC-2 lipoplexes actually promoted tumor apoptosis in vivo (Figure 6F,G), which was in line with the results of H&E staining and TUNEL assay of tumor sections (Figure 7). In addition, the results of IHC assays of tumors showed that tumors with less OC-2 positive immunostained cells had the higher proportion of apoptotic cells as well as the negative immunostained cells of Ki67, CD31 and LYVE1. Ki67, CD31 and LYVE1 are specific markers for nuclear proliferation [34,35], vascular endothelial cells [36,37] and lymphatic vessels [38,39,40], respectively, which are key for tumor metastasis and malignancy. Altogether, these findings suggested that inhibition of OC-2 expression was not conducive to tumor survival and metastasis.

### 3.7. Biodistribution and Toxicity Evaluation of cRGD-CL/pshOC-2 In Vivo

To evaluate the tumor-targeting capacity of cRGD coated liposomes, in vivo near-infrared imaging was utilized to detect the biodistribution of liposomal formulations in subcutaneous tumor-bearing mice at different time points postadministration via tail vein based on the fluorescence of DIR. In Figure 6H, within 3 h after administration, formulations in all groups were primarily enriched in the liver, while at 5 h post-injection, bare liposomes which had a smaller size than lipoplexes began to partially distribute in the tumor site regardless of their surface coating. Combined with the favored intracorporal antitumor outcome of CL/pshOC-2 lipoplexes (i.v.), we hypothesized that non-targeted CL may have acquired targeting ability in biological systems, resulting in better antitumor activity in vivo than in vitro. At 12 h post-injection, partial lipoplexes began to accumulate at tumor site, and increased with time extension. Nevertheless, the ex-vivo fluorescence image of major organs and tumors at 24 h after injection showed that liposomal formulations were primarily enriched in liver, followed by lung, spleen and kidney (Figure 6I), suggesting a potential biosafety issue of lipoplexes.

In order to evaluate the biosafety of cRGD-CL/pshOC-2 lipoplexes in vivo, organ damage was assessed by H&E staining. The results showed that, compared with the PBS treatment group, no pathological lesions were observed in the major organs of cRGD-CL/pshOC-2 lipoplexes treatment group as well as the other groups (Figure 8A). Furthermore, the results of serum biochemical indexes in each group showed that, compared with the PBS control group, no obvious adverse effects of lipoplexes on liver and renal function were found after 20 days of treatment, regardless of the route of administration (Figure 8B). These results demonstrated that cRGD-CL/pshOC-2 lipoplexes had biocompatibility in vivo without causing severe systemic toxicity. Accordingly, although lipoplexes mainly accumulated in vital organs after injection, OC-2 knockdown was unlikely to occur in normal cells due to the lack of integrin receptors in non-malignant cells. However, it is an inescapable fact that the vast majority of lipoplexes in our study were sequestered by liver and spleen where the mononuclear phagocytic system (MPS) was responsible for their degradation and elimination [41], leading to the relatively feeble accumulation of lipoplexes in tumor. Nonetheless, the weak accumulation of lipoplexes did produce OC-2 knockdown in tumor and the resultant inhibition of tumor growth and metastasis.

## 4. Discussion

The idea of DNA/RNA-based formulations (used to correct defective genes or modulate gene expression) for the treatment of breast cancer is not new and still has great appeal in the clinical market though few such drugs have passed clinical trials [42,43]. Finding an effective therapeutic gene (cancer-related genes) is a prerequisite for the success of cancer gene therapy [42,44]. OC-2 has been considered as an oncogene for its overexpressing in various types of cancers [6]. However, the role as a drug target in the treatment of breast cancer is questionable. Blocking the expression of OC-2 does not seem to yield a good outcome in breast tumor-bearing mice in a recent study [14]. In our study, RNA interference was used to investigate the role of OC-2 on the development of breast cancer. Results found that OC-2 gene silencing in MCF-7 cells led to cell apoptosis and suppressed tumor growth, suggesting the potential value of OC-2 in serving as the candidate target for breast cancer antitumor therapy. In particular, E-cadherin, which was reported to be an essential protein for tumor metastasis, was also decreased after OC-2 knockdown [25], indicating that OC-2 down-regulation was not conducive to tumor metastasis.

Another important consideration in gene therapy is the appropriate delivery system [45]. It is the key to successfully transfer nucleic acid-based drugs into tumor cells to achieve therapeutic efficacy. However, it is difficult for liposomal formulations to penetrate deeply into tumors because of the dense TME [41,46]. In our study, despite owning good properties (i.e., high transfection efficiency and selectivity for tumor cells), lipoplexes were theoretically unable to diffuse freely throughout TME because of their relative big size (i.e., lipoplexes prepared in the present study were larger than 150 nm in diameter). As a result, i.t. injection, which seemed to be the most direct and effective route of administration, failed to achieve the desired results, suggesting that i.t. injection and extravasation via size-limiting enhanced penetration and retention (EPR) effect were not conducive to the delivery of lipoplexes in tumors.

cRGD has specific affinity for the αvβ3 integrin (overexpressed in breast cancer cells and neovascular endothelial cells) [19,47] which was selected as a candidate ligand in order to enhance the accumulation and cellular uptake of cationic liposomes. However, in our study, the modification of cRGD did not lead to apparent accumulation of liposomal formulation in the tumor surface compared with the non-modified liposomes, which was inconsistent with the favorable results in vitro. Nonetheless, the limited accumulation of lipoplexes with or without cRGD modification in tumors still induced OC-2 knockdown, possibly because the PEG-modified liposomes become more dominant in the circulation and possibly because the receptor-mediated cytoplasmic action is able to penetrate the tumor efficiently. Research has demonstrated that interactions between RGD ligand and integrin receptors [20,21,22,48] or protein crona [49] coated on lipoplexes can trigger transcytosis. Transcytosis is an active receptor-mediated internalization pathway with no size limitation that mediates the transport of specific nanoparticles or macromolecules across vascular endothelial cells and tumor cells via endocytosing and secreting vesicles [50,51,52]. Therefore, tumor penetration could be achieved without being restricted by inter-endothelial gaps and densely packed TME. However, in vivo imaging showed that only a small amount of lipoplexes was enriched at the tumor site, suggesting that the lack of significant tumor regression may be attributable to the insufficient concentration of the lipoplexes reaching the tumor. In general, liposomes enter the body through intravenous injection and we prepared to obtain liposomes with positive charge, to consider whether they have an aggregation effect with serum, which was not evaluated in this study and is also to be further studied. Overall, the above results suggest that our next step could be to optimize the study of the physical properties of liposomes, how to improve the surface charge and control the vesicle size to better enable the liposomes to reach the tumor site through blood circulation.

## 5. Conclusions

Our results revealed that OC-2 knockdown was not conducive to the survival and metastasis of MCF-7 cells, indicating that OC-2 could be the potential target for breast cancer gene therapy. However, the present study could not provide strong evidence that cRGD modified or non-modified lipoplexes were penetrated into the tumor through receptor-mediated transcytosis. Thus, more studies are needed to confirm or refute this inference.

## Figures and Tables

**Figure 1 pharmaceutics-14-02157-f001:**
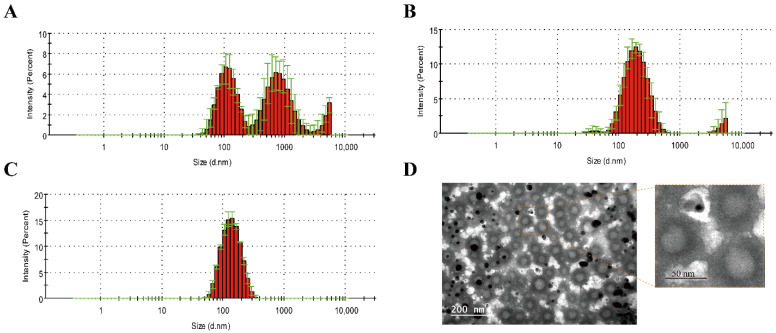
Optimization of cRGD-CL Particle Size. (**A**–**C**) The hydrodynamic diameter distribution of cRGD-CL. The lyophilized powder of cRGD-CL was detected using Zetasizer Nano ZS90 after rehydration (**A**), ultrasound (**B**) and extrusion (**C**). (**D**) Representative TEM images of cRGD-CL after extrusion. Scale bar: 200 and 50 nm.

**Figure 2 pharmaceutics-14-02157-f002:**
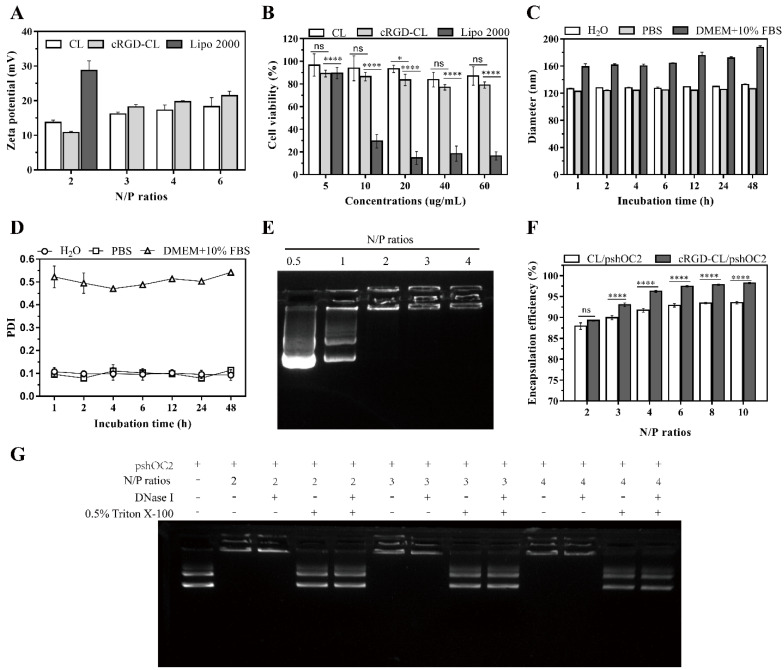
Cytotoxicity, Stability and psh*OC-2* Encapsulation Efficiency of cRGD-CL. (**A**) The zeta potential of cRGD-CL lipoplexes at different N/P ratios. (**B**) The cell viability of MCF-7 cells treated with different concentrations of CL, cRGD-CL and Lipo 2000 for 24 h, respectively. The diameter (**C**) and PDI (**D**) of cRGD-CL incubated in H_2_O, PBS and DMEM + 10% FBS within 48 h. (**E**) The 1% agarose retardation assay of cRGD-CL lipoplexes at the N/P ratios of 0.5, 1, 2, 3 and 4. (**F**) The nucleic acid encapsulation efficiency of cRGD-CL and CL with the N/P ratios at 2, 3, 4, 6, 8 and 10. (**G**) The protective effect of cRGD-CL on plasmid DNA under different N/P ratios was determined by DNase I enzymatic hydrolysis assay. Data were presented as mean ± SD, ns: No statistical difference. * *p* < 0.05, **** *p* < 0.0001.

**Figure 3 pharmaceutics-14-02157-f003:**
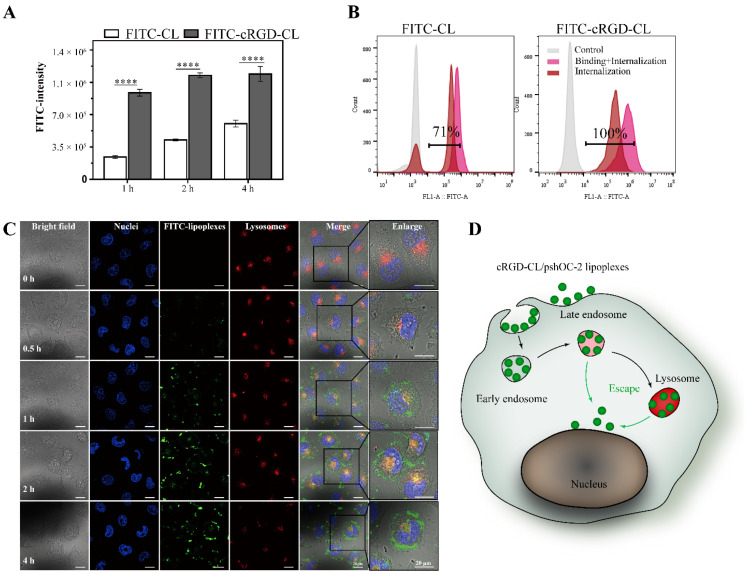
Cellular Uptake of cRGD-CL in MCF-7 Cells. (**A**) The FITC intensity of MCF-7 cells after incubation with FITC-labelled CL and cRGD-CL for 1, 2 and 4 h. (**B**) FCM results of MCF-7 cells after incubation with FITC-labelled CL and cRGD-CL for 4 h with or without 0.4% Trypan blue (**C**) Intracellular trafficking of FITC-cRGD-CL/pshOC-2 in MCF-7 cells. Cells were incubated with FITC-cRGD-CL/pshOC-2 (green) for 0.5, 1, 2 and 4 h and then labeled with Lyso-Tracker (red) and DAPI (blue) to display the lysosome and nucleus. Scale bar: 20 μm. (**D**) Schematic diagram of intracellular trafficking of FITC-cRGD-CL/pshOC-2 lipoplexes in MCF-7 cell after endocytosis. Data were presented as mean ± SD, **** *p* < 0.0001.

**Figure 4 pharmaceutics-14-02157-f004:**
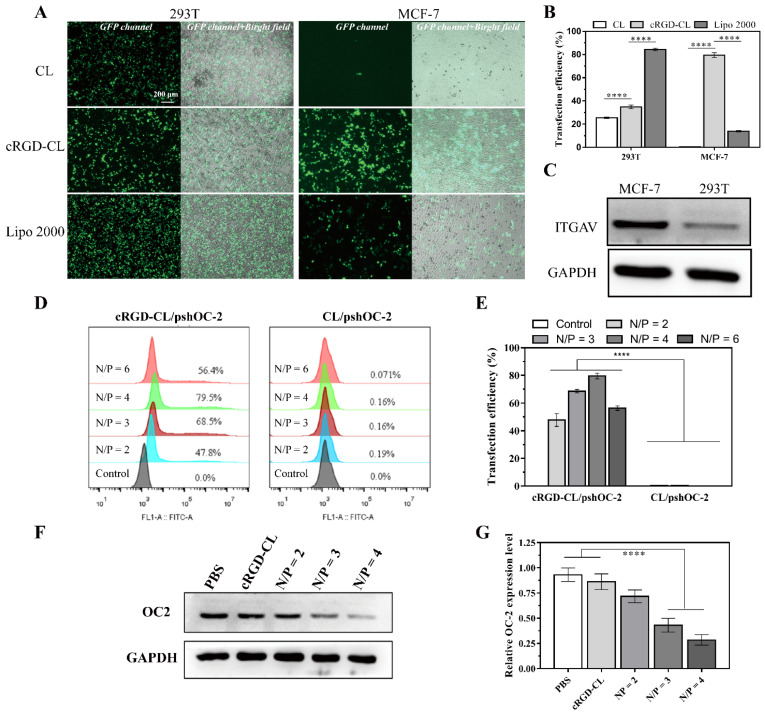
Transfection Efficiency of lipoplexes. (**A**) Fluorescence images of 293T cells and MCF-7 cells transfected with lipoplexes CL/pshOC-2, cRGD-CL/pshOC-2 and Lipo 2000/pshOC-2 for 48 h (N/P ratio = 4). Scale bar: 200 μm. (**B**) Quantitative GFP analysis of transfection efficiency. (**C**) The expression status of ITGAV in MCF-7 and 293T cells were analyzed by Western blot assay. (**D**) Histogram of GFP expression in MCF-7 cells treated with cRGD-CL/pshOC-2 and CL/ pshOC-2 (N/P ratios = 2, 3, 4 and 6). (**E**) Quantitative FCM analysis of transfection efficiency of (**D**). (**F**) The OC-2 knockdown efficiency of cRGD-CL/pshOC-2 (N/P ratios = 2, 3 and 4) in MCF-7 cells was analyzed by Western blot, followed by the quantitative analysis of relative expression (**G**). Data were presented as mean ± SD, **** *p* < 0.0001.

**Figure 5 pharmaceutics-14-02157-f005:**
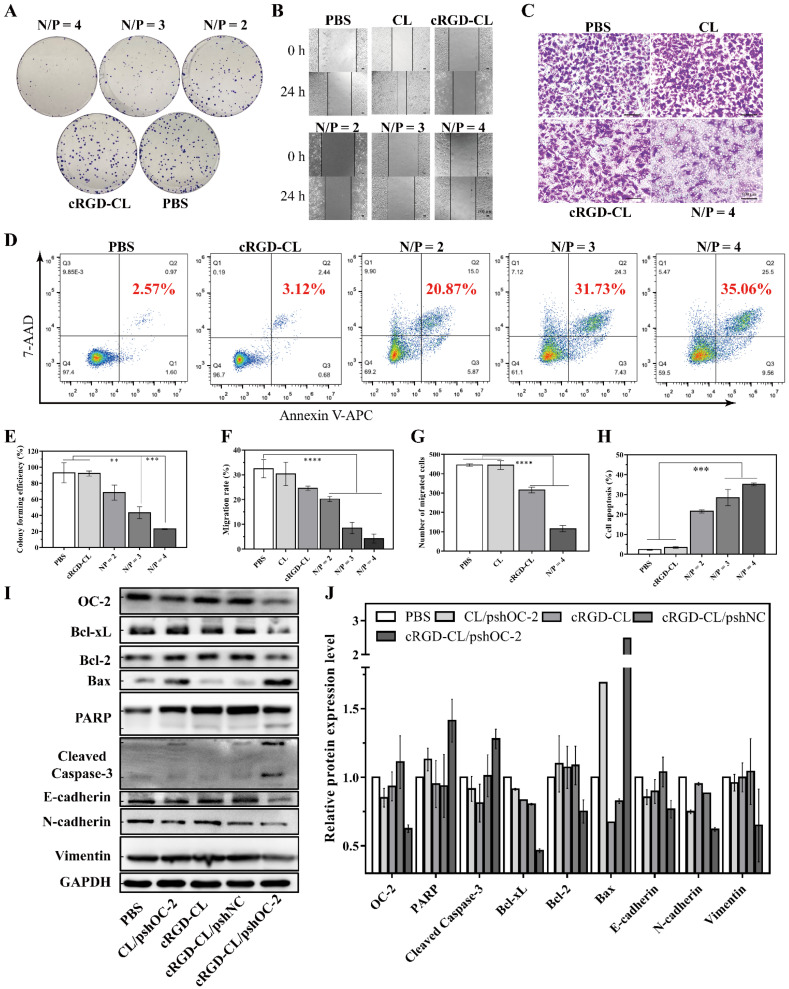
Antitumor Effects of cRGD-CL/pshOC-2 Lipoplexes on MCF-7 Cells. (**A**–**H**) MCF-7 cells were treated with cRGD-CL/pshOC-2 (N/P ratios = 2, 3 and 4). PBS served as the control. The cell proliferation, migration and apoptosis of OC-2 knockdown were evaluated by colony formation (**A**,**E**), wound-healing (**B**,**F**), Transwell (**C**,**G**) and apoptosis (**D**,**H**) assays. Scale bar: 100 μm. (**I**) The expression status of apoptosis and metastasis biomarkers were analyzed by Western blot assay, followed by the quantitative analysis of relative expression (**J**). Data were presented as mean ± SD, ** *p* < 0.01, *** *p* < 0.001, **** *p* < 0.0001.

**Figure 6 pharmaceutics-14-02157-f006:**
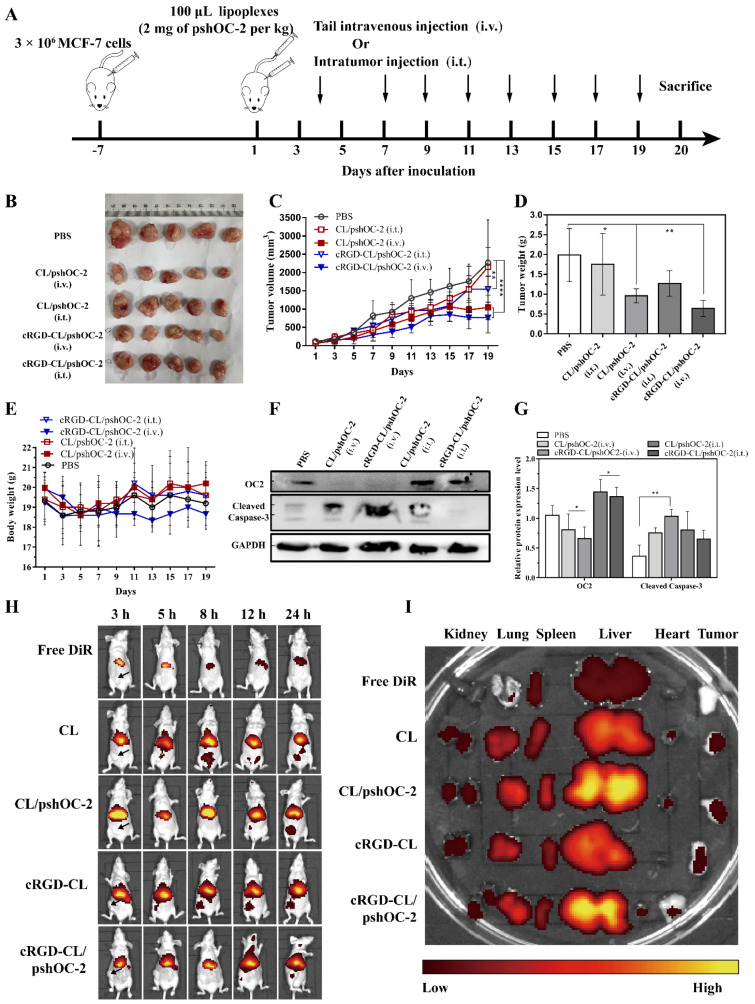
Antitumor Effects of cRGD-CL/pshOC-2 Lipoplexes on Xenografts Mouse Model. (**A**) The schematic diagram of administration of liposomal formulation to xenografts mouse model. MCF-7 cells were subcutaneously injected into the armpit of BALB/c-nude mice (n = 5 per group). CL/pshOC-2 and cRGD-CL/pshOC-2 (2 mg of pshOC2 per kg) lipoplexes were intravenously or intratumorally injected when the xenografts reached approximately 100 mm^3^. Mice were sacrificed after 9 times treatment. (**B**) The photograph of striped tumors and statistical analyses of (**C**) tumor volume, (**D**) tumor weight and (**E**) body weight of each group. (**F**) The expression status of OC-2 and Cleaved Caspase 3 in tumor samples were analyzed by Western blot assay, with (**G**) statistical analysis. (**H**–**I**) Biodistribution of cRGD-CL/pshOC-2 lipoplexes in tumor-bearing mice. The tumor-bearing mice were intravenously injected with free DiR, CL, CL/pshOC-2, cRGD-CL and cRGD-CL/pshOC-2, respectively. The fluorescence intensity was detected by IVIS Imaging System. (**H**) In vivo imaging showing the distribution and accumulation of cRGD-CL/pshOC-2 at 3, 5, 8, 12 and 24 h. (**I**) Major organs and tumors were imaged 24 h after injection. Data were presented as mean ± SD, ns: No statistical difference. * *p* < 0.05, ** *p* < 0.01, **** *p* < 0.0001.

**Figure 7 pharmaceutics-14-02157-f007:**
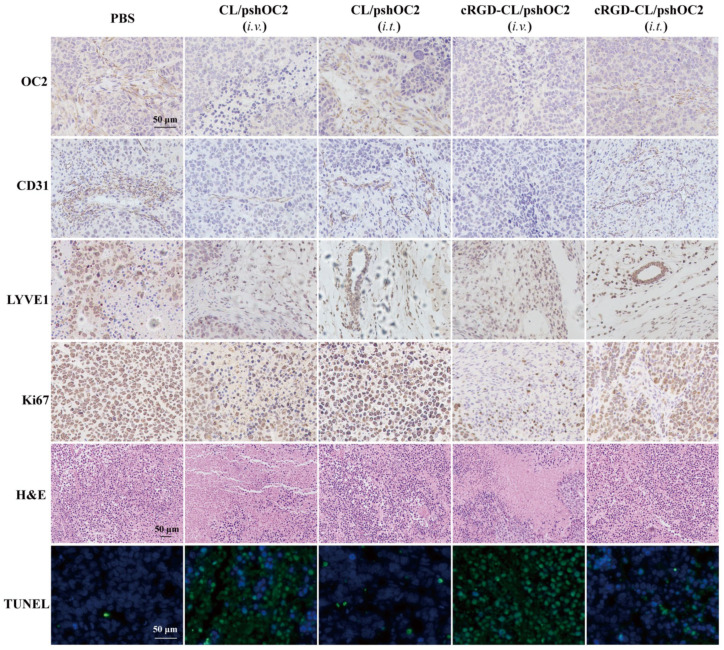
Representative Images of H&E, IHC and TUNEL Assays. Tumor tissues treated with CL/pshOC-2 and cRGD-CL/pshOC-2 were fixed and sectioned. H&E staining and the protein expression of Ki67 for tumor growth, CD31 for tumor angiogenesis, LYVE1 for tumor lymphangiogenesis and TUNEL for tumor cell apoptosis. Scale bar: 50 μm.

**Figure 8 pharmaceutics-14-02157-f008:**
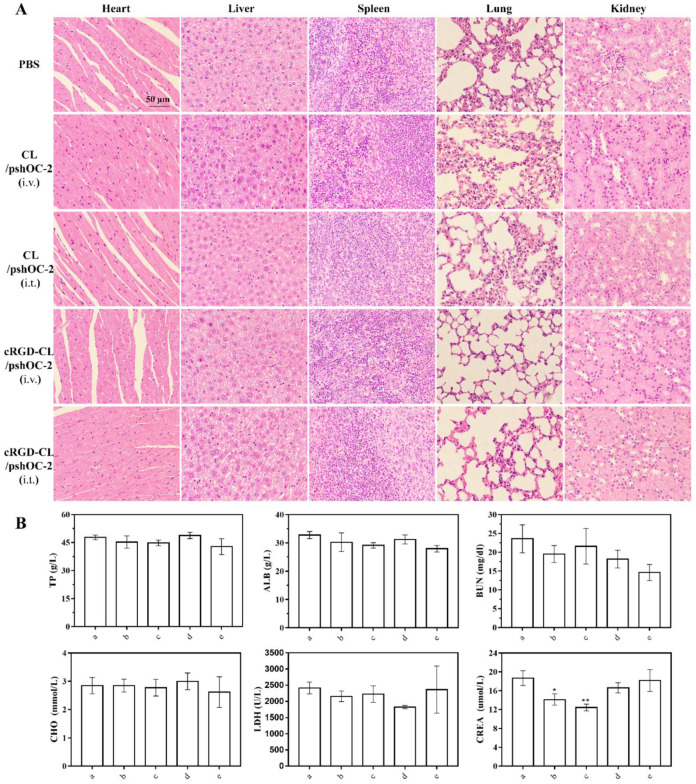
Analyzes of Organs Pathology and Serum Biochemistry. (**A**) H&E staining of major organs from each group. Scale bar: 50 μm. (**B**) Serum levels of blood markers. Liver function: albumin (ALB); Renal function: blood urea nitrogen (BUN) and creatinine (CREA); Blood lipid: cholesterol (CHO); Myocardium: lactate dehydrogenase (LDH); Others: total protein (TP). a: PBS, b: CL/pshOC-2 (i.v.), c: CL/pshOC-2 (i.t.), d: cRGD-CL/pshOC-2 (i.v.), e: cRGD-CL/pshOC-2 (i.t.). Data were presented as means ± SD, * *p* < 0.05, ** *p* < 0.01.

**Table 1 pharmaceutics-14-02157-t001:** Particle Size and PDI of cRGD-CL after Extrusion or Ultrasound.

Items	Untreated	Extrusion	Ultrasound
Size (nm)	235.5 ± 9.6	126.6 ± 0.92	192.2 ± 6.01
PDI (polydispersity index)	0.802 ± 0.05	0.108 ± 0.02	0.355 ± 0.05

**Table 2 pharmaceutics-14-02157-t002:** Characterization of pshOC-2-delivered CLs.

Formulations	Composition	Size (nm)	Zeta-Potential (mV)	PDI
cRGD-CL	DOTAP: cholesterol: PEG2k-DMG: cRGD-PEG2k-DSPE, 50:42:2:6, molar ratios	125 ± 0.918	35 ± 0.368	0.108 ± 0.015
cRGD-CL lipoplexes	cRGD-CL/pshOC-2 (N/P = 4)	150 ± 1.020	19 ± 0.249	0.132 ± 0.009
CL	DOTAP: cholesterol: PEG2k-DMG, 50:42:8, molar ratios	146 ± 1.250	25 ± 1.970	0.161 ± 0.011
CL lipoplexes	CL/pshOC-2 (N/P = 4)	183 ± 2.110	17 ± 1.230	0.167 ± 0.004

## Data Availability

Not applicable.

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
