# Peer review of "Preparation of DSPE-PEG-cRGD Modified Cationic Liposomes for Delivery of OC-2 shRNA and The Antitumor Effects on Breast Cancer"

_pharmaceutics, 2022, doi:10.3390/pharmaceutics14102157_

Round 1

Reviewer 1 Report

1. Authors need to discuss why they chose shRNA rather than siRNA in nonviral formulation.

2. Authors need to intracellular imaging data of FITC-CL to compare with FITC-cRGD-CL

3. Authors should replace blot data of cleaved caspase-3 in Fig.5I. The quality is very low and not suitable for publication. 

4. Authors need to add introduction of cationic liposomes. Please refer the following paper; https://doi.org/10.1016/j.ajps.2013.07.009.

Author Response

Response to Reviewer 1 Comments

Point 1 : Authors need to discuss why they chose shRNA rather than siRNA in nonviral formulation.

Response 1 : We agree and appreciate this suggestion. We have described this issue in detail in the revised manuscript, as follows:

(Line 61 - 64): Compared with shRNA, the electrostatic interaction between siRNA and liposomes is not controlled, which may lead to excessive particle size and poor stability. Another potential problem is the low siRNA encapsulation rate, which exposes siRNA to po-tential degradation by enzymes before entering cells[16].

Point 2 : Authors need to intracellular imaging data of FITC-CL to compare with FITC-cRGD-CL

Response 2 : Since cRGD-CL is the focus of our study and its destination after entering the cell is of interest, we studied its endosomal escape and did not focus on FITC-CL.

Point 3 : Authors should replace blot data of cleaved caspase-3 in Fig.5I. The quality is very low and not suitable for publication.

Response 3 : We are sorry for the inconvenience caused by the unclear pictures. The high-resolution version has been uploaded and we recommend you download it for viewing.

Point 4 : Authors need to add introduction of cationic liposomes. Please refer the following paper; htts://doi.org/10.1016/j. ajps 2013.0700(.).

Response 4 : Thank you for the good suggestion. We have added a more detailed description of cationic liposomes in the introduction, and also, we have cited the suggested reference. In the revised manuscript, the part has been added as follows:

(Line 57 - 70): The transfection efficiency of cationic liposomes is lower than that of viral vectors, but they are considered to be more valuable gene vectors for research because of their excellent biosafety, transfection efficiency and biocompatibility; cationic liposomes can be loaded with different nucleic acid molecules, plasmids and siRNAs are more com-mon[15]. Compared with shRNA, the electrostatic interaction between siRNA and liposomes is not controlled, which may lead to excessive particle size and poor stability. Another potential problem is the low siRNA encapsulation rate, which exposes siRNA to potential degradation by enzymes before entering cells[16]. Cationic liposomes (CLs) are positively charged subspherical vesicles, mainly composed of cationic liposomes and neutral. The positive charge on the surface of cationic liposomes promotes contact with negatively charged cell membranes[17], but on the other hand, they also bind to serum proteins and form larger polymers that do not easily enter the cells, so it is imperative to modify the surface of cationic liposomes, and the common way is PEG modification, which also reduces the toxicity of cationic liposomes[18].

Reviewer 2 Report

The work is interesting, with many methods that follow each other logically and with good results. The interpretation of the results and the conclusions are good. Before publishing the article, some corrections are necessary: The manuscript needs to be checked by a native English speaker. There are many words that are used incorrectly (probably due to translation) and some phrases are difficult to read. For example, cRGD tripeptide is flourished / biodistribution and biosafety in mice were valued / reared in the Animal Center of Jinan University / hydraulic diameter / against the degradation of endonuclease...

Also, the abstract must be verified. The second sentence has no verb. Figure 2 is missing. Page 13, row 430: What is "dysopsonions"?

Author Response

Response to Reviewer 2 Comments

Point 1 : The work is interesting, with many methods that follow each other logcally and with good results. The interpretation of the results and the conclusions are good. Before publishing the article, some corrections are necessary: The manuscript needs to be checked by a native English speaker. There are many words that are used incorrectly (probably due to translation) and some phrases are difficult to read. For example, cRGD tripeptide is flourished / biodistribution andbiosafety in mice were valued / reared in the Animal Center of Jinan University / hydraulic diameter / against the degradation of endonuclease.

Response 1 : Thank you very much for your good suggestions. We have carefully checked and improved the English writing in the revised manuscript. Please see if the revised version met the English presentation standard. Several English writing problems pointed out by the reviewers in the manuscript were revised separately. As follows:

(Line 28 - 41) : Breast cancer is the second leading cause of mortality among women, with the highest incidence worldwide in 2020 [1,2]. Endocrine therapy and chemotherapy are the primary treatments for breast cancer, but their therapeutic efficacy is limited and occa-sionally unsatisfactory due to the high tumor heterogeneity and drug resistance[3,4]. Gene therapy is one of the most promising and effective therapeutic approaches in breast cancer treatment, which has attracted widespread attention and expectations, and the discovery of new therapeutic targets and delivery systems are of great im-portance[5].

(Line 73 - 77) : As integrin αvβ3 is  upregulated in tumor and angiogenic endothelial cells[19,20], it is widely used in the recognition of cRGD tripeptide in nucleic acid delivery systems [21].

(Line 84 - 86) : The effects of cRGD-CL/pshOC-2 liposome-mediated OC-2 knockdown on colony for-mation, wound healing and migration of MCF-7 cells were investigated, and their an-titumor activity, biodistribution and biosafety were evaluated in mice.

(Line 112 - 113) : Female 4-week-old BALB/c nude mice were purchased from Beijing Huafukang Biological Co. Ltd (China).

(Line 133 - 135) : The hydrodynamic diameter, polydispersity (PDI) and zeta potential of the opti-mized blank liposomes (cRGD-CL or CL) were measured using a Zetasizer Nano ZS90 (Malvern Instruments).

(Line 148 - 153) : In addition, the protective effect of cRGD-CL on pshOC-2 (0.3 μg) was determined by DNase I enzymatic hydrolysis assay. The cRGD-CL/pshOC-2 lipoplexes with N/P ratios of 2, 3 and 4 were interacted with DNase I for 10 min and then DNase I was in-activated in a water bath at 60 °C for 5 min, and the liposomes were disrupted with 0.1% TritonX-100 to release their contents. Finally, all samples were subjected to 1% agarose gel electrophoresis.

Point 2 : Also, the abstract must be verified. The second sentence has no verb.

Response 2 : Thank you for the good advice. The sentence has been added with a verb to make the sentence flow correctly. The second sentence of the abstract has been modified to “ the cyclic Arg-Gly-Asp (cRGD) modified cationic liposomes (cRGD-CL) were designed for targeted delivery of ONECUT2 (OC-2) shRNA (pshOC-2) to breast cancer cells”.

Point 3 : Figure 2 is missing.

Response 3 : Thank you so much for your careful check. We have added the figure of Figure 2 into the manuscript.

Point 4 : Page 13, row 430: What is “dysopsonions”?

Response 4 : We apologize for the misspelling of "dysopsonins" as "dysopsonions." We have corrected it and added a further explanation to help readers understand.

(Lines 460 - Lines 464) : Liposomal drugs are coated with plasma proteins once injected intravenously into the physiological environment [29]. Dysopsonins are one such plasma proteins adsorbing on the surface of foreign particles, which endows the particles with the ability to evade capture by mononuclear phagocyte system (MPS) due to their few affinities for the cell surface [30-32].

References

[29] Gao H, He Q. The interaction of nanoparticles with plasma proteins and the consequent influence on nanoparticles behavior. Expert Opin Drug Deliv. 2014 Mar;11(3):409-20. doi: 10.1517/17425247.2014.877442.

[30] Papini E, Tavano R, Mancin F. Opsonins and Dysopsonins of Nanoparticles: Facts, Concepts, and Methodological Guidelines. Front Immunol. 2020 Oct 12;11:567365. doi: 10.3389/fimmu.2020.567365.

[31] Lu X, Xu P, Ding HM, Yu YS, Huo D, Ma YQ. Tailoring the component of protein corona via simple chemistry. Nat Commun. 2019 Oct 4;10(1):4520. doi: 10.1038/s41467-019-12470-5.

[32] Absolom DR. Opsonins and dysopsonins: an overview. Methods Enzymol. 1986;132:281-318. doi: 10.1016/s0076-6879(86)32015-9. 

Round 2

Reviewer 1 Report

The authors have addressed the comments I made in my previous review.